

# Physiologic effects of surgical masking in children versus adults

J Patrick Brooks[1,2], Jill Layman[1] and Jessica Willis[3]

[1] School of Anesthesia, Missouri State University, Springfield, Missouri, United States
[2] Department of Biomedical Sciences, Missouri State University, Springfield, Missouri, United States
[3] RStats Institute, Missouri State University, Springfield, Missouri, USA

## ABSTRACT

**Background:** Surgical masks remain a focal part of the CDC guidelines to decrease COVID-19 transmission. Evidence refuting significant effects of masking on ventilation is mostly limited to small studies, with a paucity of studies on children, and none comparing children to adults.

**Methods:** A total of 119 subjects were enrolled (71 adults, 49 children) in a prospective interventional study with each subject serving as their own mask-free control. End tidal $CO_2$ (ETCO2), inspired $CO_2$ (ICO2), and respiratory rate were measured by nasal cannula attached to an anesthesia machine D-fend module. Pulse oximetry and heart rate were also followed. After the mask-free period, an ASTM Level 3 disposable surgical mask was donned and 15 min of mask-worn data were collected.

**Results:** A steady state was confirmed for ETCO2 and ICO2 over the masked period, and mean ICO2 levels rose significantly ($p < 0.001$) after masking in all age groups. The increase in ICO2 for the 2- to 7-year-old group of 4.11 mmHg (3.23–4.99), was significantly higher ($p < 0.001$) than the final ΔICO2 levels for both the 7- to 14-year-old group, 2.45 mmHg (1.79–3.12), and adults, 1.47 mmHg (1.18–1.76). For the pediatric group there was a negative, significant correlation between age and ΔICO2, r = −0.49, $p < 0.001$. Masking resulted in a statistically significant ($p < 0.01$) rise in ETCO2 levels of 1.30 mmHg in adults and 1.36 mmHg in children. The final respective ETCO2 levels, 34.35 (33.55–35.15) and 35.07 (34.13–36.01), remained within normal limits. Pulse oximetry, heart rate, and respiratory rate were not significantly affected.

**Discussion:** The physiology of mechanical dead space is discussed, including the inverse relationship of subject age *vs* ICO2. The methodology and results are compared to previously published studies which detracted from the physiologic safety of surgical masking.

**Conclusions:** The wearing of a surgical mask results in a statistically significant rise in ICO2 and a smaller rise in ETCO2. Because ETCO2 and other variables remain well within normal limits, these changes are clinically insignificant.

Corresponding author
J Patrick Brooks,
patrickbrooks@missouristate.edu

## INTRODUCTION

Extensive evidence accumulated in 2020 revealing that surgical masks were effective in mitigating the spread of respiratory viral pathogens and specifically SARS-CoV-2, and these masks were similar in effectiveness to N95 filtering facemask respirators (*Violante & Violante, 2020*; *Bartoszko et al., 2020*). By the end of the summer of 2020, over 30 US states and territories had some form of mandatory statewide mask mandate due to the COVID-19 pandemic (*Strand et al., 2022*). With the advent of mandates, public pushback evolved to include concerns that surgical masks elevated carbon dioxide levels or impaired oxygenation in the wearers, especially in children. Research into that topic was limited, with physiologic studies on mask-wearing pediatric subjects especially rare, and none compared the relative effects between children and adults. Yet in 2021 and 2022, four peer reviewed publications claimed that masking was physiologically harmful, with one evaluating the effect of masking on children (*Christakis & Fontanarosa, 2021*; *Elbl et al., 2021*; *Kisielinski et al., 2021*; *Akhondi et al., 2022*).

The primary aim of this study is to compare the physiological effects of surgical masks on children and adults in the largest study on this topic to include physiologic endpoints. End tidal $CO_2$ will be measured as the primary endpoint, with secondary endpoints of inspired $CO_2$, respiratory rate, oxygen saturation, and heart rate. As the first to include broad age ranges, this study will investigate the relationship between age and mechanical dead space effects. Because a tidal volume brings in a large volume of fresh air, it is hypothesized that end tidal $CO_2$ levels will remain within the normal range for all age groups. Because the dead space to tidal volume ratio is higher in pediatric patients (*Pearsall & Feldman, 2014*; *Numa & Newth, 1996*), it is hypothesized that masking will result in higher inspired $CO_2$ levels in younger patients when compared to adults. An additional goal of this study is to compare results to the prior publications which detracted from the physiological safety of surgical masking.

## METHODS

This is a prospective interventional study with each subject serving as their own mask-free control. Subjects were sequentially assigned to the experimental arm of the study after donning a disposable surgical mask. The protocol was approved by the Missouri State University Institutional Review Board (approval number FY2022-36) and was listed on the ClinicalTrials.gov website (identifier: NCT05114993). Healthy volunteer adults and parents with children were recruited for the study. No remuneration or other direct benefits were provided. Inclusion criteria included age 2 to 14 years (inclusive) or 18 to 80 years (inclusive). Exclusion criteria included significant cardiopulmonary disease, symptoms of active respiratory infection, intolerance to wearing a nasal canula, or intolerance to wearing a surgical mask. A preliminary power determination recommended adult and pediatric group sizes of 38 participants each. Interest in participation was high, therefore enrollment of subjects continued until all interested parties were able to participate, reaching group numbers comparable to or exceeding prior studies on this topic. Written consent was obtained from adults and parents of study subjects. Children

aged seven and older provided written assent, with verbal assent obtained from younger subjects.

The study took place at the Missouri State University School of Anesthesia Simulation Operating Room. Three separate anesthesia machine monitors allowed for evaluation of up to three subjects simultaneously, allowing children and parents to participate seated in the same room. An anesthesia machine D-fend module measured end tidal carbon dioxide (ETCO2), inspired carbon dioxide (ICO2), and respiratory rate (RR) by way of nasal canulae. Oxygen saturation (SpO2) and heart rate (HR) were followed by pulse oximetry. These five physiologic variables were recorded each minute over a 5-min control period while subjects were unmasked. The subjects were then assisted in donning a DemeTECH ASTM Level 3 Surgical Disposable Mask. Appropriate fit was confirmed, using masks of either regular or small size. These three-layer masks provide >98% filtration efficiency and >98% sub-micron particulate filtration efficiency at 0.1 micron. The masks fully covered the mouth and nose, with the nose wire formed around the nose and cheek to close any gap. Each subject used a new nasal canula and a new surgical mask for the study. The five physiologic variables (ETCO2, ICO2, RR, SpO2, HR) were then measured on masked participants each minute for 15 min. This duration was chosen based on a pilot study which showed stability of ETCO2 and ICO2 levels over a 15-min masked period.

## Statistics

Looking to detect a rise of ETCO2 of 1.0 mmHg ($f = 0.19$) as statistically significant ($p < 0.05$) for our primary endpoint, an *a priori* power analysis was conducted with the G*Power statistical program using a within-subject design, alpha of 0.05, and power of 0.80. Results showed that 38 participants for each age group (pediatrics and adults) were required to achieve a significant effect with adequate power. Secondary endpoints included the remaining physiologic variables of ICO2, RR, SpO2, and HR.

Averages for each of the five physiologic variables were calculated over four time periods: 5-min mask free average, first 5-min masked average, second 5-min masked average, and last 5-min masked average. ΔICO2 values were calculated as the rise in mean ICO2 for each 5-min masked period compared to the mask free mean ICO2 level. One-Way Repeated Measures ANOVA was performed to examine differences in the four time periods' mean ETCO2, ICO2, ΔICO2, RR, SPO2, and HR values. *Post hoc* paired samples *t*-tests with a Bonferroni correction were conducted to detect differences between time period mean values, with a difference considered significant at $p < 0.050$.

Statistical analyses were first performed on the pediatric and adult groups' data, with ETCO2 levels remaining in the normal range as described in the Results section. Because changes in mechanical deadspace are expected to have a larger effect on the youngest subjects, *post hoc* subgroup analysis further investigated the effects relative to age. Subgroups were created by dividing the pediatric group at the chronological midpoint, for subgroup sizes of 20 subjects aged 2–7 years and 28 subjects aged 7–14. A Pearson's correlation coefficient was also computed to assess the linear relationship between age and ΔICO2 in the main groups.

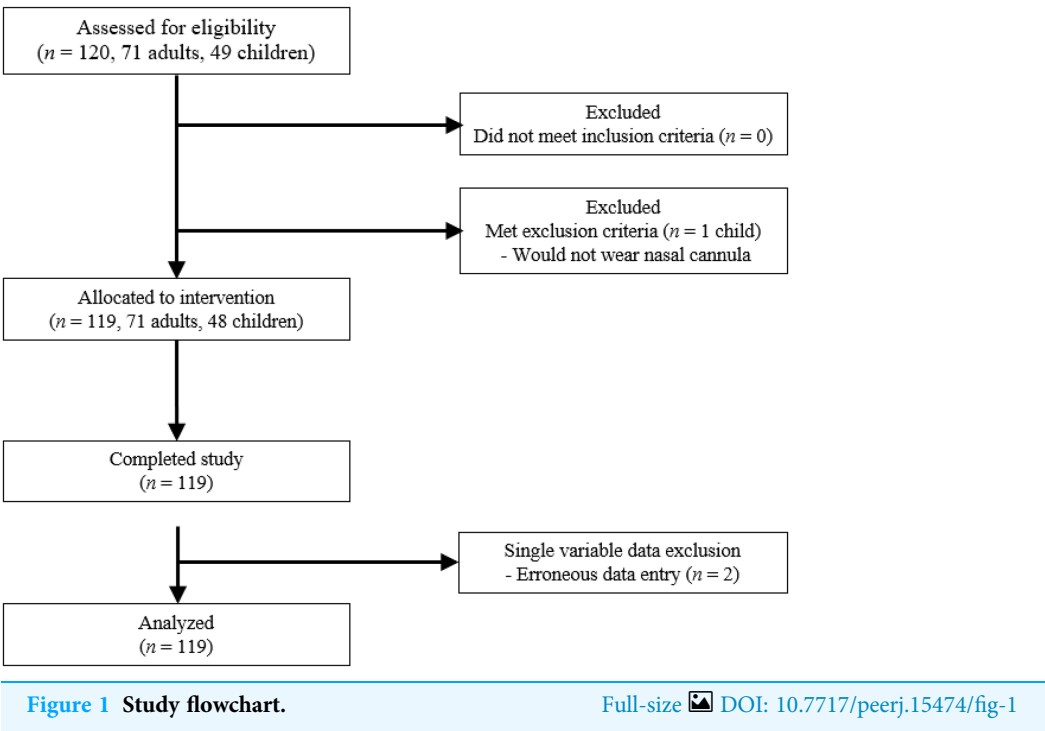

**Figure 1 Study flowchart.** 

In preliminary analyses, the resulting data for the five physiologic variables were screened to assess accuracy, missing data, outliers, and the violation of the normality assumption and homogeneity assumption. Out of 11,900 data points, spurious data entries were identified in one subject's heart rate and one subject's ICO2, and the data from these two participants were removed from the respective analyses. The visual inspection of standardized histograms revealed the assumption for normality was met, with slight skewing for some of the physiological markers. Lastly, the homogeneity assumption was met for all the parametric analyses except for the SPO2 and RR physiological markers for the adult participants and ETCO2 and ICO2 for the pediatric participants. Therefore, a Huynh-Feldt or Greenhouse-Geisser correction was used when appropriate.

## RESULTS

The study was performed between November 16, 2021 and January 27, 2022, with 119 participants completing the study. No potential participant required exclusion due to health. One 2-year-old subject was unwilling to wear the nasal canula and was excluded from the study (Fig. 1). The adult group was approximately 80% female and included 70 participants, with ages ranging from 18 to 66 and a mean age of 35.0 years. The pediatric group was 60% female and included 48 participants, with ages ranging from 2 to 14 and a mean age of 8.3 years. For subgroup analysis of the youngest participants, 20 of these pediatric participants were between the ages of 2 and 7, with a mean age of 5.1 years.

The 15-min masked period was confirmed statistically as a steady state. Pairwise comparisons of ETCO2 levels between each 5-min masked period revealed no significant differences in either age group ($p > 0.155$), Similarly, no significant differences were noted

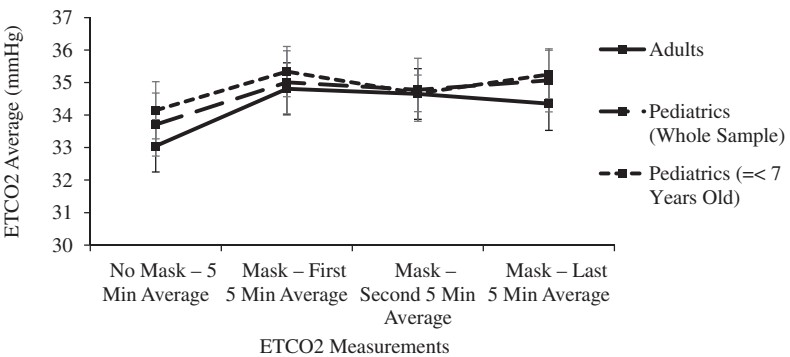

**Figure 2 Changes in adult and pediatric end tidal carbon dioxide (ETCO2) as a function of mask and time.**

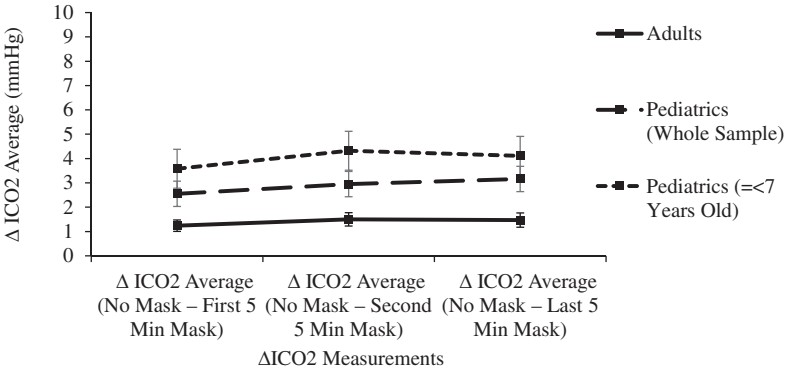

**Figure 3 Changes in adult and pediatric delta inspired carbon dioxide (delta CO2) as a function of mask and time.**

in comparisons between each masked period's ICO2 levels ($p > 0.320$) or ΔICO2 levels ($p > 0.826$) (Figs. 2 and 3).

In both the adult and pediatric age groups, mean ETCO2 levels were significantly ($p < 0.01$) increased in comparisons between any of the 5-min masked periods and the unmasked control period (Table 1 and Fig. 2). Despite these small ETCO2 increases, 1.30 mmHg in adults and 1.36 mmHg in children, the ETCO2 levels in both age groups remained within normal limits during the masked periods: adult levels of 33.05 mmHg (32.27–33.38) increased to 34.35 mmHg (33.55–35.15), and pediatric levels of 33.71 mmHg (32.77–34.65) increased to 35.07 mmHg (34.13–36.01).

A subgroup analysis was performed for mean ETCO2 levels in 20 pediatric patients ranging in age from 2 to 7 years old. After 15 min of masking, the ETCO2 levels rose by 1.1 mmHg but this change did not reach statistical significance after controlling for Type I error ($p = 0.102$). The ETCO2 levels remained within the normal range with a value of 35.25 mmHg (33.78–36.72). For this subgroup of youngest subjects, the respiratory rate significantly ($p < 0.001$) increased after masking from 18.48 (16.93–20.03) to 21.63 (19.55–23.71), yet the respiratory rate remained normal throughout the masked evaluation period. The respiratory rate improved during the last 5 min of masking, resulting in final

**Table 1 Descriptive statistics and ANOVA results of the five physiologic markers for adults and pediatric patients.**

| Physiological marker | Adult patients | | Pediatric patients (Whole sample) | | Pediatric patients (<7 Years old) | |
|---|---|---|---|---|---|---|
| | Mean (95% CI) | *p*-value | Mean (95% CI) | *p*-value | Mean (95% CI) | *p*-value |
| End-tidal (ETCO2) | | <0.001 | | 0.001 | | 0.035 |
| No mask—5 min average | 33.05 [32.27–33.38] | | 33.71 [32.77–34.65] | | 34.15 [32.43–35.87] | |
| Mask—first 5 min average | 34.81 [34.03–35.59] | | 35.01 [34.07–35.95] | | 35.34 [33.83–36.85] | |
| Mask—second 5 min average | 34.65 [33.89–35.41] | | 34.78 [33.84–35.72] | | 34.67 [33.55–35.79] | |
| Mask—last 5 min average | 34.35 [33.55–35.15] | | 35.07 [34.13–36.01] | | 35.25 [33.78–36.72] | |
| ΔInspired carbon dioxide (ICO2) | | <0.001 | | <0.001 | | <0.001 |
| No mask—first 5 min average | 1.24 [1.00–1.48] | | 2.55 [2.03–3.07] | | 3.58 [2.78–4.38] | |
| No mask—second 5 min average | 1.50 [1.23–1.77] | | 2.95 [2.43–3.46] | | 4.32 [3.52–5.12] | |
| No mask—last 5 min average | 1.47 [1.18–1.76] | | 3.16 [2.64–3.67] | | 4.11 [3.31–4.91] | |
| Respiratory rate (RR) | | 0.130 | | 0.057 | | <0.001 |
| No mask—5 min average | 14.09 [13.46–14.72] | | 16.85 [15.56–18.14] | | 18.48 [16.93–20.03] | |
| Mask—first 5 min average | 13.41 [12.76–14.06] | | 17.71 [16.42–19.00] | | 20.80 [18.78–22.82] | |
| Mask—second 5 min average | 13.70 [12.84–14.56] | | 18.15 [16.86–19.44] | | 21.63 [19.55–23.71] | |
| Mask—last 5 min average | 13.96 [13.14–14.78] | | 17.45 [16.16–18.74] | | 19.97 [16.87–23.07] | |
| Pulse oximetry (SPO2) | | 0.506 | | 0.391 | | 0.05 |
| No mask—5 min average | 97.72 [97.35- 98.09] | | 97.80 [97.51–98.09] | | 97.40 [96.81–97.99] | |
| Mask—first 5 min average | 97.64 [97.29–97.99] | | 98.00 [97.71–98.29] | | 98.12 [97.71–98.53] | |
| Mask—second 5 min average | 97.84 [97.55–98.13] | | 98.02 [97.73–98.31] | | 98.04 [97.69–98.39] | |
| Mask—last 5 min average | 97.79 [97.50–98.08] | | 97.89 [97.60–98.18] | | 97.82 [97.37–98.27] | |
| Heart rate (HR) | | 0.088 | | 0.410 | | 0.025* |
| No mask—5 min average | 77.47 [74.67–80.27] | | 94.86 [91.84–97.88] | | 96.70 [91.82–101.58] | |
| Mask—first 5 min average | 76.46 [73.79–79.13] | | 95.50 [92.48–98.52] | | 98.30 [93.20–103.4] | |
| Mask—second 5 min average | 76.88 [74.19–79.57] | | 95.84 [92.82–98.86] | | 99.77 [95.48–104.6] | |
| Mask—last 5 min average | 76.68 [74.05–79.31] | | 94.92 [91.90–97.94] | | 98.66 [94.03–103.29] | |

Note:
* After controlling for a Type I error, ETCO2 and HR are no longer significant for the pediatric patients <= 7 years old.

masked respiratory rates having no significant difference in comparison to the unmasked rates in these youngest subjects ($p$ = 0.241): 18.48 (16.93–20.03) to 19.97 (16.87–23.07) (Table 1, Figs. 2 and 4).

In both the adult and pediatric groups, mean ICO2 levels were significantly ($p$ < 0.001) increased in comparisons between the unmasked control period and any of the 5-min masked periods. In comparisons between age groups, the Final ΔICO2 was significantly higher ($p$ < 0.001) in children, 3.16 mmHg (2.64–3.67), compared to adults, 1.47 mmHg (1.18–1.76) (Table 1 and Fig. 2). With ICO2 levels showing the most variability related to age, subgroup analyses of pediatric subjects were performed to allow comparisons between three age groups: children 2 to 7 years of age ($n$ = 20), children >7 to 14 years of age ($n$ = 28), and adults ($n$ = 71). Prior to masking, there were no significant differences in the mean ICO2 levels between the three age groups ($p$ > 0.3). However, the final 5-min masked period ΔICO2 for the 2- to 7-year-old group of 4.11 mmHg (3.23–4.99), was significantly higher ($p$ < 0.001) than the final ΔICO2 levels for both the 7- to 14-year-old group, 2.45

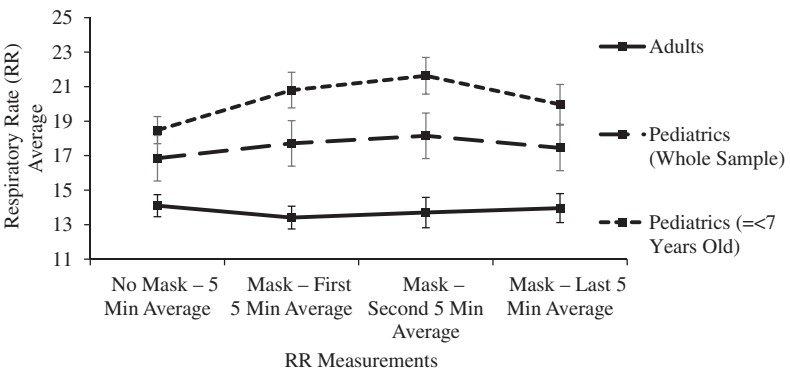

**Figure 4 Changes in adult and pediatric respiratory rate (RR) as a function of mask and time.**

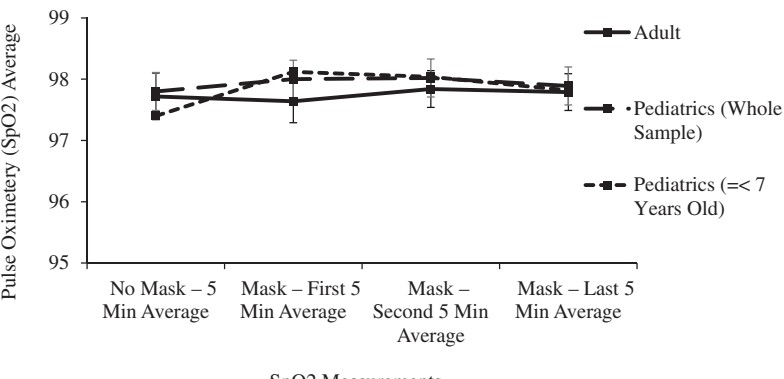

**Figure 5 Changes in adult and pediatric pulse oximetry (SpO2) as a function of mask and time.**

mmHg (1.79–3.12), and adults, 1.47 mmHg (1.18–1.76). Additionally, the final 5-min masked period ΔICO2 was significantly higher for the 7- to 14-year-old group compared to adults ($p = 0.018$) (Fig. 3).

These exploratory analyses should be taken with caution as the sample size for the pediatric subgroups compared to adults was small. Therefore, a Pearson's correlation coefficient was computed to assess the linear relationship between age and ΔICO2. For the adult group, there was no linear correlation between ΔICO2 and age, r = −0.12, $p = 0.332$. For the pediatric group, however, there was a negative, significant correlation between the two variables, r = −0.49, $p < 0.001$.

In the main two groups of adult and children, there was no significant change in respiratory rate, pulse oximetry, or heart rate after masking (Table 1 and Figs. 4–6).

## DISCUSSION

Surgical-type facemasks have been in use for over one hundred years, with the first major study performed by Doust in 1918 evaluating their use in the prevention of respiratory pathogen transmission (*Doust & Lyon, 1918*). Airborne transmission of SARS-Cov-2 in highly contagious aerosols has been established as the dominant route, making the wearing

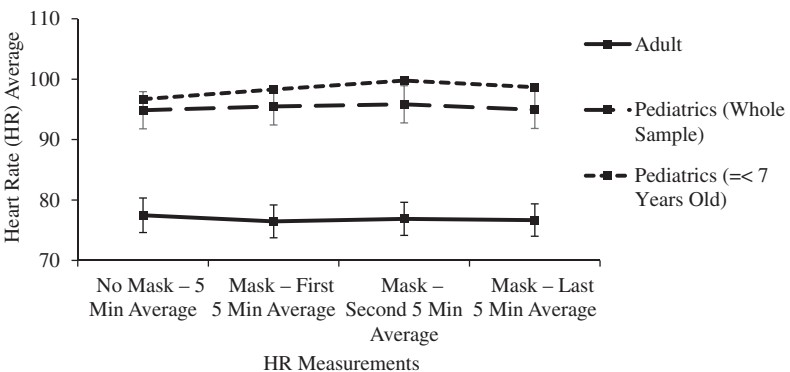

**Figure 6 Changes in adult and pediatric heart rate (HR) as a function of mask and time.**

of face masks in public one of the most effective means to prevent transmission (*Zhang et al., 2020*; *Ju, Boisvert & Zuo, 2021*). The CDC and other national agencies have emphasized the importance of masking as a means to decrease the transmission of SARS-CoV-2 by respiratory droplet transmission. During the study period, the American Academy of Pediatrics stated it "strongly recommends that anyone over the age of 2, regardless of vaccination status, wear a well-fitting face mask when in public" (https://www.aap.org, *Accessed July 20, 2021*). At the time of submission of this manuscript, the peak of the COVD-19 pandemic had passed, yet the CDC continued to include recommendations for surgical masking for high risk individuals, those with confirmed or suspected exposure to an infected person for 10 days, all infected persons when around others at home and in public for 10 days, and all citizens during high COVID-19 community levels (*Massetti et al., 2022*). FDA guidelines state that N95 respirators are not designed or recommended for children (https://www.fda.gov, *Accessed January 6, 2023*), therefore this study focused only on surgical masks.

The first 90 years of surgical mask use passed with minimal concerns regarding ventilation or oxygenation. Assessments of carbon dioxide levels in masked participants were quite rare prior to the COVID-19 pandemic. A 2012 study evaluated twenty adult participants unmasked on a treadmill for 1 h followed by 1 h while masked, finding clinically insignificant increases in respiratory rate and transcutaneous carbon dioxide levels (*Roberge, Kim & Benson, 2012*). During the COVID-19 pandemic, concerns about carbon dioxide levels behind masks prompted further investigation. Small studies in masked pilots in an altitude chamber (*Dattel et al., 2020*) and masked, ambulating COPD patients found no major changes in ETCO2 levels or oxygenation (*Samannan et al., 2021*).

Physiologic assessments of pediatric participants wearing surgical masks are even less common, with none identified prior to the COVID-19 pandemic. A 2021 study evaluated triple layer surgical face masks on 47 children, including many younger than 2 years of age (*Lubrano et al., 2021*). The masked period was 30 min, yet data points were only recorded every 15 min. Masking did not affect ETCO2 levels at rest or after 12 min of ambulation. Clinical ICO2 monitoring was not used in that 2021 study, nor any other pediatric study of surgical masking to date. Later in 2021, a peer-reviewed publication claimed that

mask-wearing was dangerous for children, due to their inhaled air having higher CO2 levels than allowed by factory environmental standards (*Christakis & Fontanarosa, 2021*). The study focused only on gas measurements behind the masks, with no assessments of any PaCO2 analogue. Shortly thereafter, the study was retracted due to "study methodology, including concerns about the applicability of the device used for assessment of carbon dioxide levels in this study setting, and whether the measurements obtained accurately represented carbon dioxide content in inhaled air, as well as issues related to the validity of the study conclusions (*Christakis & Fontanarosa, 2021*)".

A later 2021 study focused on an ETCO2 endpoint and suggested masks caused physiologic harm, although the study model did not include any live subjects. Using a lung simulator and intubation head, the simulation resulted in an average ETCO2 increase of 17.4 mmHg (*Elbl et al., 2021*). No attempts were made to explain how a century of mask wearing health care providers have been able to tolerate CO2 levels that, by these estimates, would reach 57 mmHg.

Publications like the above retracted study fueled increasing claims about physiologic harm caused by mask wearing. Multiple lawsuits against Ohio school districts (*Dayton Daily News, 2021*) have cited an additional 2021 publication: a review of multiple adult studies assessing the effects of mask wearing on CO2 levels and other physiologic measurements (*Kisielinski et al., 2021*). The authors argue that these studies show a proven effect of masks increasing CO2 levels and lowering blood oxygen saturation and therefore "Long-term disease-relevant consequences of masks are to be expected" (*Kisielinski et al., 2021*). Further inspection of the CO2 measurements in the cited primary source manuscripts reveals that only one study evaluated surgical masks exclusively, and it revealed a small, clinically insignificant rise in transcutaneous CO2 despite exercise (*Roberge, Kim & Benson, 2012*). Another study evaluated working subjects who wore surgical masks or N95 respirators, showing no clinically significant increase in CO2 levels (*Georgi et al., 2020*). Most studies in this review exclusively evaluated N95 respirators which appear to cause a slightly higher increase in CO2 levels than surgical masks, yet in healthy subjects the changes in CO2 levels were still referred to as "clinically insignificant" or "within normal limits" (*Goh et al., 2019*; *Bharatendu et al., 2020*; *Roberge et al., 2010*; *Rebmann, Carrico & Wang, 2013*). This remained true when working in an N95 respirator (*Roberge, Kim & Powell, 2014*) or pregnant and exercising in an N95 respirator (*Roberge, Kim & Powell, 2014*; *Tong et al., 2016*). To achieve clinically significant elevations in CO2 beyond normal levels, it required exercising to the point of exhaustion in an N95 respirator (*Epstein et al., 2021*), or mask wearing in patients with severe COPD or acute exacerbations of COPD (*Kyung et al., 2020*; *Mo et al., 2020*).

With the paucity of studies on the topic of surgical masking, small sample sizes focusing on one age group, and variable recorded physiological data, this study was designed to be the largest of its kind assessing the effect of surgical masks on both end tidal CO2 levels and inspired CO2 levels. It is the first study to compare both end tidal and inspired CO2 levels in masked children, and the first to compare the effects of mask dead space in children *vs* adults. Measurement of PaCO2 levels is invasive and impractical in volunteer subjects, but sidestream ETCO2 monitoring by nasal cannula has proved accurate as an assessment of

PaCO2 in adults and children (*Barton & Wang, 1994*; *Abramo et al., 1995*). This study used a D-Fend module for sidestream ETCO2 monitoring, the format which has been standard of care on anesthesia machines for the assessment of ventilation under general anesthetic for over 25 years and under moderate or deep sedation for over a decade. Multiple nasal cannula designs have been shown to provide accurate ETCO2 waveforms, with the highest accuracies obtained with the patients breathing room air as in our study (*Ebert et al., 2015*). The recorded levels may be 2 to 3.5 mmHg lower than arterial blood gas or capillary CO2 levels (*Barton & Wang, 1994*; *Butterworth, Mackey & Wasnick, 2018*), yet this noninvasive technology is especially useful for following trends over any length of time. ICO2 is routinely displayed on anesthesia machines with the ETCO2, yet there has been minimal research about the utility of ICO2 monitoring. Although ICO2 monitoring is not within the anesthesia standard of care at this time, a rise in ICO2 is accepted as a clinical assessment of CO2 rebreathing (*Barash et al., 2017*). ICO2 monitoring has been suggested as an important metric to follow in sedated, spontaneously breathing patients to avoid adverse respiratory events from increased dead space ventilation under operating room drapes (*McHugh, 2019*).

Surgical masks have a pore size of around 20 micrometers, with CO2 molecules measuring 0.32 nanometers and O2 molecules measuring even smaller. Even triple layer surgical masks like the models used in this study have high breathability, as measured by the low differential pressure of <5 mm H20/cm$^2$. Neither oxygen nor carbon dioxide will be obstructed in its flow across a surgical mask, yet some amount of expired carbon dioxide may remain behind the mask in the form of a mechanical dead space at the end of the expired breath. A significant increase in dead space decreases the effective minute ventilation and raises the PaCO2 and therefore the ETCO2. Each inspired breath has a slightly higher CO2 concentration compared to baseline, confirmed in this study by the small increase in ICO2 of 3.16 mmHg in children and half that value in adults (Table 1 and Fig. 3). This leads to a small but statistically significant increase in ETCO2, yet even the pediatric group's post-mask ETCO2 levels of 35.07 (34.13–36.01) mmHg remain well within the normal range (34–42 mmHg) (*Butterworth, Mackey & Wasnick, 2018*) because the absolute rise in ETCO2 of only 1.30 mmHg in adults and 1.36 mmHg in children is clinically quite small (Table 1 and Fig. 2).

Increases in mechanical dead space (or apparatus dead space) are of particular importance in pediatric patients because of their larger dead space to tidal volume ratio (*Pearsall & Feldman, 2014*). Anatomic dead space in an adult is 2.2 ml/kg, yet because of the relatively larger head size of infants and children, anatomic dead space increases with decreasing age, exceeding 3 ml/kg in early infancy (*Numa & Newth, 1996*). In this study, participants' ICO2 levels prior to masking were not statistically different between age groups. Nonetheless, the ΔICO2 was the focus of the statistical evaluation (rather than the total ICO2) since the rise in ICO2 is specific to the deadspace effects of masking. Ten to fifteen minutes after donning a surgical mask, a stepwise increase in ΔICO2 was noted in comparisons of the three age groups of adults, older children, and younger children, thus confirming the greater influence of mechanical deadspace in younger participants (Fig. 3). This inverse relationship between age and the effect of mechanical deadspace is further

confirmed in a linear fashion by a significant negative Pearson correlation coefficient for the pediatric subjects ($r = -0.49$, $p < 0.001$). No such correlation was seen in the adult subjects, whose large tidal volume to deadspace ratio can easily tolerate small additions of mechanical deadspace. Although an inverse relationship between age and masked $ICO_2$ levels was confirmed, the increased $ICO_2$ did not have clinically significant effects on the $ETCO_2$ even in the youngest subgroup, since $ETCO_2$ rose by only 1.1 mmHg and remained in the normal range.

Clinically, it is well known that an increase in mechanical dead space can have significant effects on $PaCO_2$ levels, especially in the youngest of pediatric patients. In a study of infants and young children, adding a heat and moisture exchanger (HME) into the ventilation circuit increased the $PaCO_2$ inversely proportional to weight and age. In healthy pediatric patients weighing more than 25 kg, however, the additional 22 ml of dead space from the HME had no effect (*Kwon, 2012*). Supraglottic airway devices have larger internal volumes than endotracheal tubes, and the use of these devices may affect ventilation in some instances. In a study of children under age 6 comparing these two devices, however, $ETCO_2$ levels were not significantly different (*Goenaga-Diaz et al., 2021*). In the smallest of children, or those with cardiopulmonary disease, the addition of mechanical dead space can have clinically significant effects. Masking is not recommended for children under the age of 2. Surgical masks also have excellent breathability, whereas ventilator circuitry does not allow any escape of $CO_2$ or oxygen.

The retracted 2021 study (*Christakis & Fontanarosa, 2021*) and a similar 2022 study (*Akhondi et al., 2022*) detract from the safety of masking by using $CO_2$ meters to focus on gas levels behind masks, comparing those levels to standards meant for a surrounding environment. Claims are made that clinical symptoms of hypercapnia will ensue, while avoiding any measurement of a $PaCO_2$ analogue or oxygen saturation. The small mechanical dead space behind a surgical mask with high breathability should be compared to tidal volumes of 5–8 ml/kg for children and roughly 500 ml for an adult, which ensure adequate ventilation to prevent hypercapnia. Some of these flawed publications remain in print, including a second study by Wallace which followed his retraction, still free of any endpoint assessment of $PaCO_2$ and no mention of this critical omission in the study limitations (*Walach et al., 2022*). These arguments in the detracting literature, focusing only on $CO_2$ levels behind masks without attempting to measure a physiologic endpoint, would appear to be in bad faith.

The argument in the 2021 Kisielinski review article (*Kisielinski et al., 2021*), that any increase in $CO_2$ level is potentially harmful even while remaining well within the reference range, has no basis in clinical practice or in reputable publications. All authors of that review's primary source articles (and other studies reviewed in this manuscript) discount these small fluctuations of normal $CO_2$ levels as clinically insignificant. To evaluate the standard of care, a review of over 300,000 patients whose ventilation was managed under general anesthetic calculated the mean $ETCO_2$ as 35 (33.0–38.0) mmHg (*Akkermans et al., 2019*). Our post-mask $CO_2$ measurements are at the midpoint of this range as measured by the same technology. The same review also confirms that the medical professionals who manage $ETCO_2$ levels most attentively are not concerned with small fluctuations, since

there was wide variation in acceptable levels and an increasing tolerance of ETCO2 levels over 45 mmHg. The trend in acceptance of higher CO2 levels is related to the growing body of evidence that high normal or even slightly elevated CO2 levels are beneficial (*Akkermans et al., 2019*; *Way & Hill, 2011*). While hypocapnia has long been known to reduce cerebral blood flow, normal or mildly elevated CO2 levels improve cerebral perfusion and are associated with improved postoperative cognitive function. There are several other known benefits of avoiding hypocapnia: increased subcutaneous oxygen tension, protection against organ injury, reduced postoperative infection rates, improved recovery time from general anesthetic, and improved tissue oxygenation through increased cardiac output and increased oxygen offloading. ETCO2 levels at the midpoint of the reference range are not pathologic.

Limitations of the study include a masked observation period limited to 15 min, and the evaluation of subjects only at rest. Longer observation times and the effects of masking during exercise have been reported in other smaller studies, as noted above. In this study, ETCO2 and ICO2 levels were significantly increased from baseline within the first 5 min of masking, and pairwise comparisons between each 5-min masked period thereafter confirmed the ETCO2 and ICO2 levels were at equilibrium (Table 1, Figs. 2 and 3). Within this manuscript's clinical references, 15 min was also chosen as the acceptable time period between dead space manipulations and arterial blood gas measurements (*Goenaga-Diaz et al., 2021*). With this stability initially noted in the pilot study, longer observation times were avoided as they would have decreased volunteer participation. Tidal volume was not measured in this study or other studies on this topic, yet it is telling that the final masked respiratory rates were not significantly increased compared to the control, mask-free period. In an environment that traps a significant volume of CO2, such as beneath surgical drapes for ophthalmologic surgery, the respiratory rate does rise considerably, and it does not improve until the mechanical dead space is eliminated (*Schlager, 1999*). This study's youngest subjects did increase their respiratory rate early after masking, yet the 15-min study period was long enough to confirm a decrease in the respiratory rate to the resting level, providing further evidence of the adequate observation period in this study.

## CONCLUSIONS

Compared to a mask free period, wearing an ATSM 3 triple layer surgical mask resulted in a small increase in ICO2 consistent with the mechanical deadspace behind the mask. The rise in ICO2 levels varied inversely with subject age, reflecting the known increase in dead space to tidal volume ratio of the youngest subjects. ETCO2 increased in all age groups by a lesser amount, but most importantly, ETCO2 levels remained in the normal range even in the youngest subject subgroup. These small, clinically insignificant changes in ETCO2 were not enough to prompt a sustained increase in respiratory rate. Oxygen saturation and heart rate were unaffected by surgical masking.

During pandemics current and future, the wearing of surgical masks may be encouraged in adults and children over age 2 without concerns of the effects of carbon dioxide retention or impaired oxygenation.

## ACKNOWLEDGEMENTS

The authors would like to express our appreciation to our student research group: Reagan Stange, Ashlyn Spinabella, Ashlynn Harmon, Breanna Skinner, Caleb Dodd, Carla Casteñeda, Glory Ehie, Kaitlyn Miller, Kaity Kuhnert, Kayla Kline, and Krusha Bhakta.

### Funding

The authors received no funding for this work.

### Competing Interests

The authors declare that they have no competing interests.

### Author Contributions

- J. Patrick Brooks conceived and designed the experiments, performed the experiments, analyzed the data, prepared figures and/or tables, authored or reviewed drafts of the article, and approved the final draft.
- Jill Layman conceived and designed the experiments, performed the experiments, analyzed the data, prepared figures and/or tables, authored or reviewed drafts of the article, and approved the final draft.
- Jessica Willis conceived and designed the experiments, analyzed the data, prepared figures and/or tables, authored or reviewed drafts of the article, and approved the final draft.

### Human Ethics

The following information was supplied relating to ethical approvals (*i.e.*, approving body and any reference numbers):

Missouri State University Institutional Review Board.

### Data Availability

The raw data is available in the Supplemental File.

### Clinical Trial Registration

The following information was supplied regarding Clinical Trial registration:

ClinicalTrials.gov NCT05114993.

### Supplemental Information

Supplemental information for this article can be found online at http://dx.doi.org/10.7717/peerj.15474#supplemental-information.

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
