# Peer review of "Physiologic effects of surgical masking in children versus adults"

_PeerJ, doi:10.7717/peerj.15474_

## Round 0.1 · original submission · Major Revisions

The authors have appealed the rejection decision and I have agreed to change the decision to a revision decision, to give them the opportunity to respond to the comments in a formal revision and rebuttal. I ask that the authors respond to all the comments from the reviewers and properly address those in the revised version.

· Appeal

Appeal


· · Academic Editor

Reject

Based on the critical issues pointed out by reviewer 3, We are not able to accept your manuscript.

·

Basic reporting

This is a well organized, well written and well presented study of the physiologic impacts of surgical mask use in pediatric and adult subjects.

Experimental design

The study is prospective, with each subject serving as their own control. The authors are thorough in providing the background used to support the reliability and validity of the study design.

-The authors should explain where subjects were identified/recruited from. Were these patients in a particular clinic? Or healthy volunteers?

-The authors should better explain why enrollment so far exceeded the anticipated participation outlined by the power calculation. In particular, did participants receive remuneration or any other direct benefits from participation?

-As the power calculation was performed to assess for significant change in End tidal C02, the authors should consider designating ETC02 as the primary endpoint of the study, and the remainder as secondary endpoints.

-Presentation of the reason for and approach to the subgroup analyses in the methods section would strengthen this publication.

Validity of the findings

The authors rightly identify and address the two main vulnerabilities of the study - the short duration of monitoring and that monitoring was done only at rest. Additional studies addressing these issues would be helpful to confirm generalizability of the findings to real world use, but the manuscript remains reliable and conclusions are supported by the data presented.

Reviewer 2 ·

Basic reporting

The article has been written in an unambiguous and clear way. It shows professional English throughout its sections. Background to the area of research has been compiled with sufficient citations from the relevant literature. It manifests a professional article structure, figures, tables and raw data. It is self-contained with the relevant results to the hypothesis under test.

Experimental design

This article manifests original primary research within aims and scope of the journal.
Research question are well defined, relevant & meaningful. It is stated how research fills an identified knowledge gap.

Rigorous investigation was performed to a high technical & ethical standard.

Methods have been described with sufficient detail & information to replicate.

Validity of the findings

The article presented novel results which are of high impact. Meaningful replication encouraged where rationale & benefit to literature is clearly stated.

All underlying data have been provided; they are robust, statistically sound, & controlled.

Conclusions are well stated, linked to original research question & limited to supporting results.

Additional comments

This research article presents high impact novel results. It has been compiled very well. All the sections are well written in a professional scientific way.
Though, the experimental sample taken was not so large, the impact of the study design can not be neglected and the results presented are a step forward to answer common myths spread elsewhere behind negative impact of face masks.

·

Basic reporting

no comment

Experimental design

Relative difference cannot be ignored. Just because ETCO2 etc. remain within normal limits, we can’t conclude that everything is fine. The significant fall does imply importance. It can lead to decreased productivity and efficiency that we can’t ignore. Cut-off values for physiological parameters can’t always be assessed as a binomial variables of ‘within range’, and ‘outside range’. A consistent deviation from earlier values may also have clinical significance.
On top of that, the authors have considered a rise of 1mm of EtCO2 as significant in the methodology. Given that they have found a greater rise, how can they ignore this in the conclusion?
Again, a value greater than one is not considered significant here, “Although an inverse relationship between age and masked ICO2 levels was confirmed, the increased ICO2 did not have clinically significant effects on the ETCO2 even in the youngest subgroup, since their ETCO2 rose by only 1.1 mmHg and remained in the normal range.”


“Enrollment of adult and pediatric subjects continued until all interested
78 parties were able to participate, reaching group numbers beyond those suggested by a
79 preliminary power determination. ” The authors have not mentioned the calculated sample size. This is necessary when making such a statement.
Moreover, the pre-specified power is also not mentioned


“For subgroup analysis of the youngest
129 participants, twenty of these pediatric participants were between the ages of 2 and 7, with a mean
130 age of 5.1 years.”
What was the basis for keeping the age of seven years as a barrier? Moreover, this subgroup analysis wasn’t specified in the methodology. Was it pre-specified or post-hoc? If the latter, it should be mentioned as such.


“This duration was chosen based on
96 a pilot study which showed stability of ETCO2 and ICO2 levels over a 15-minute masked
97 period.” - The pilot study should be referenced here so that others can check it.

Validity of the findings

The authors have also not discussed critical relevant information discussed in some recent articles like this:
Walach H, Traindl H, Prentice J, Weikl R, Diemer A, Kappes A, Hockertz S. Carbon dioxide rises beyond acceptable safety levels in children under nose and mouth covering: Results of an experimental measurement study in healthy children. Environ Res. 2022 Sep;212(Pt D):113564. doi: 10.1016/j.envres.2022.113564. Epub 2022 May 28. PMID: 35636467; PMCID: PMC9142210.

---

## Round 0.2 · accepted · Accept

Your manuscript is acceptable for publication

Reviewer 2 ·

Basic reporting

The article has been written in an unambiguous and clear way. It shows professional English throughout its sections. Background to the area of research has been compiled with sufficient citations from the relevant literature. It manifests a professional article structure, figures, tables and raw data. It is self-contained with the relevant results to the hypothesis under test.

Experimental design

This article manifests original research which lie within the aims and scope of the journal. Research question is defined well. Methods have been revised very well and can be replicated by other scientists now working in the relevant fields.

Validity of the findings

All novel results have been reported well in the frame/scope of the study. Now, the conclusions are also well stated by the authors.

Additional comments

Not required.